# Cryptic diploid lineage of *Betula ermanii* at its southern boundary populations in Japan

**Takaki Aihara**[1], **Kyoko Araki**[2], **Yoshihiko Tsumura**[3]*

**1** Graduate School of Life and Environmental Sciences, University of Tsukuba, Tsukuba, Ibaraki, Japan,
**2** Graduate School of Science and Technology, University of Tsukuba, Tsukuba, Ibaraki, Japan, **3** Faculty of
Life and Environmental Sciences, University of Tsukuba, Tsukuba, Ibaraki, Japan

* tsumura.yoshihiko.ke@u.tsukuba.ac.jp

Cryptic diploid lineage of *Betula ermanii* at its
southern boundary populations in Japan. PLoS
ONE 19(7): e0307023. https://doi.org/10.1371/
journal.pone.0307023

University, INDIA

**Data Availability Statement:** All relevant data are
within the manuscript and its Supporting
Information files.

**Funding:** This work was supported by: the JSPS
KAKENHI program, Recipient: YT, Grant Numbers:

## Abstract

Polyploidy is thought to enable species diversification and adaptation to extreme environments. Resolving the ecological differences between a taxon's ploidy levels would therefore provide important insights into local adaptation and speciation. The genus *Betula* includes many polyploids, but estimates of their phylogenetic relationships and evolutionary history are uncertain because of cryptic lineages and species. As one of the southern boundary populations of *Betula ermanii* in Japan has been shown to have distinctive genetic characteristics and traits, the differences in ploidy levels between three southern boundary and various other Japanese *B. ermanii* populations were investigated using flow cytometry. Leaf and seed morphologies were also compared. Apart from individuals in southern boundary populations, all those sampled were tetraploid. Individuals from the southern boundary populations were mostly diploid, apart from a few from lower altitude Shikoku populations, which were tetraploid. Leaf and seed morphologies differed between tetraploids and diploids. Diploid individuals were characterized by leaves with a heart-shaped base and many leaf teeth, and seeds with relatively longer wings. The diploid populations could be considered a cryptic relict lineage of *B. ermanii*, and there is a possibility that this lineage is a diploid ancestor of *B. ermanii* and a relict population of the Sohayaki element. Further investigation of the Japanese *Betula* phylogenetic relationships would enable an informed discussion of taxonomic revisions.

## Introduction

It has been hypothesized that polyploidy, having multiple sets of chromosomes as a consequence of whole-genome duplication, is a driver for species diversification and can enable plant species to adapt to extreme environments [1]. Polyploidy can mask deleterious recessive mutations even in isolated and bottlenecked populations, mainly because of its increased genetic variation [2]. Furthermore, especially in allopolyploids, which originate from hybrids of diploid progenitors, the fixing of divergent parental genomes and duplicated copies of important or essential genes can be advantageous in more diverse and stressful environments [3–5]. For example, [6] demonstrated that the number of high-level and recently evolved

21H04732 and 24H12345, URL: https://www.jsps.go.jp/j-grantsinaid/; JST Next Generation Researchers Challenging Research program, Recipient: TA, Grant Number: JPMJSP2124, URL: https://www.jst.go.jp/jisedai/. The funders had no role in study design, data collection and analysis, decision to publish, or preparation of the manuscript.

**Competing interests:** The authors have declared that no competing interests exist.

polyploids increases with latitude within the Arctic. Similarly, polyploids have larger cell volumes than diploids because of whole-genome duplication, and larger stomata and xylem vessel diameters, hence they display a higher resistance to water stress and can occupy a wider distribution under drier conditions [7, 8]. Identifying differences in traits and distributions between ploidy levels would therefore provide important insights into adaptive evolution and species diversification.

The genus *Betula* includes about 65 species and subspecies in the Northern Hemisphere [9], and is characterized by frequent inter-specific hybridization [10, 11]. Nearly 60% of *Betula* species are polyploids [12], and some *Betula* species include lineages that display different levels of ploidy, such as *B. papyrifera* (2×, 3×, 5× and 6×) [13], *B. pendula* (2× and 4×) [14], *B. chinensis* (6× and 8×) [9] and *B. dahurica* (6× and 8×) [12]. Recent research has resolved some of the phylogenetic relationships for the genus *Betula* using genome-wide markers, and identified the progenitors of certain polyploids [15]. However, uncertainties remain regarding its phylogenetic history. Section *Costatae* of the genus *Betula* includes species distributed across east Asia. As there are only two diploid species within this section, *B. costata* and *B. ashburneri*, all the polyploid progenitors within *Costatae* are assigned to them [15]. However, the spatial distribution of these two species is such that a hybrid zone seems unlikely: *B. costata* is distributed across northern and north-eastern China and the Russian Far East [9], whereas *B. ashburneri*, a relatively newly discovered species, is distributed disjunctively in south-east Tibet and may have distributions in central China, such as Shaanxi province [16]. One of the earlier studies aimed to resolve the taxonomic uncertainty in *Costatae* and considered *B. ashburneri* and the diploid lineage of two species in the Qinling Mountains, China, were to be the same species [17]. In addition, they described a new diploid species, *B. buggsii*, found in the Qinling Mountains [17]. Furthermore, a cryptic distribution of *B. costata* in Japan has been identified [18, 19]. The existence of more unknown lineages and/or species within section *Costatae* are therefore suspected.

*Betula ermanii* belongs to section *Costatae* and is a tetraploid species found in cool and snowy environments across eastern Russia, northern China, Korea and Japan. This species has a wide range of distributions and tree forms, from a canopy tree up to 20 m tall in deciduous forests to a shrub in harsh alpine timberline [9]. Then, studying the genetic and phenotypic variation including ploidy level of *B. ermanii* could be key to understand the adaptive evolution of tree species. While now combined as one species, *B. ermanii*, called 'dakekamba' in Japanese, in the past separate species were recognized in each region of Japan [20], probably because of their broad distribution range and phenotypic plasticity. For example, *B. ermanii* populations on Shikoku Island have been described as *B. shikokiana* NAKAI, called 'shikoku-dakekamba' in Japanese [21–23], and individuals with red bark and a heart-shaped leaf base have been described as *B. ermanii* var. *subcordata* KOIDZ, called 'aka-kamba' [20, 24, 25]. One of the populations at the species' southern boundary has been described as having distinctive genetic characteristics and traits and suffering from inbreeding depression [26], which is likely to be masked by the gene redundancy of polyploidy [2]. In addition, in the same regions as the southern boundary populations of *B. ermanii* on the Kii Peninsula and Shikoku, there are many relict endemic species called 'Sohayaki elements' [27] and distinct lineages of some coniferous species such as *Picea jezoensis*, *Abies homolepis* and *Thuja standishii* [28–30]. To clarify the presence of cryptic lineages distributed across these regions, we therefore surveyed three southern boundary populations of *B. ermanii*: one population on the Kii Peninsula and two populations on Shikoku Island. For comparison, we also collected samples from populations in other regions of Japan. We investigated the ploidy level of these populations using flow cytometry, and quantified differences in their morphology, and present a discussion of the importance newly found lineages of *B. ermanii*.

## Materials and methods

### Study area and sampling

This study focused on *B. ermanii* populations at the species' southern boundary in Japan, on the Kii Peninsula and Shikoku Island (Fig 1). These populations exist at about 1700–1900 a.s.l., and are isolated from other *B. ermanii* populations in northern Japan by their high altitude location and relatively small size (Fig 1).

To estimate the ploidy level, leaf samples of *B. ermanii* were collected from three southern populations, Shaka-gatake (SHK), Tsurugi-san (TRG) and Ishizuchi-san (ISZ) (Figs 1 and 2), in September 2022. For each population, the ploidy level was estimated from 10 individuals scattered along the elevation (S1 Table). For comparison, leaf samples were also collected from 11 populations elsewhere across Japan (Fig 1; Table 1): Uryu (URU), Akkeshi (AKS), Hakkoda (HKD), Goyo-san (GYS), Choukai-san (CKS), Bandai-san (BDS), Mikuni-touge (MKT), Yatsu-gatake (YGT), Alps-west (APW), Nougouhaku-san (NGH), and Alps-south (APS). For YGT, the leaf samples were collected from two wild individuals in September 2022. For URU, AKS, HKD, GYS, CKS, BDS, MKT, APW, NGH and APS, which their saplings were planted in common garden populations as stated in [26], the leaf samples were collected from two saplings per population in September 2022 or June 2023, (Table 1). Based on the RAD-seq analysis [26], it was observed that SHK, one of the southern boundary populations, could potentially be diploid. In this study, to confirm this observation, we conducted flow cytometry. Consequently, with the exception of three southern boundary populations, we sampled two samples from each population as controls.

To compare the morphology between *B. ermanii* populations, we collected leaf and seed samples from seven populations, URU, HKD, BDS, YGT, SHK, TRG and ISZ, in August and

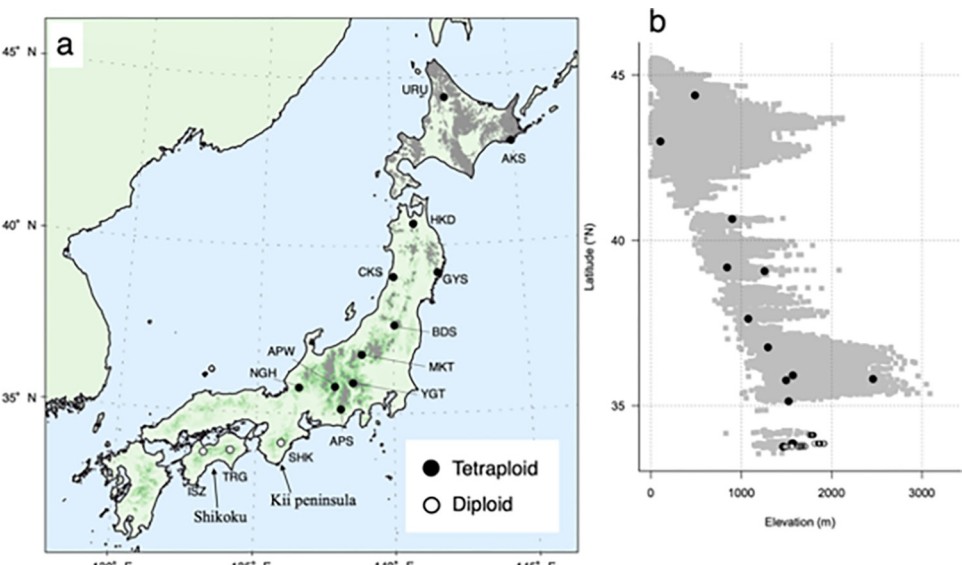

**Fig 1. The distribution of the study populations of *Betula ermanii*.** The location of the study populations (a) and their vertical distribution (b). Black circles indicate tetraploid populations; white circles indicate diploid populations; gray shading indicates habitat suitable for *B. ermanii* as predicted by niche modeling [26]. The 14 populations were: URU, Uryu; AKS, Akkeshi; HKD, Hakkoda; GYS, Goyo-san; CKS, Choukai-san; BDS, Bandai-san; MKT, Mikuni-touge; YGT, Yatsu-gatake; APW, Alps-west; NGH, Nougouhaku-san; APS, Alps-south; SHK, Shaka-gatake; TRG, Tsurugi-san; ISZ, Ishizuchi-san. The shapefiles of land boundary were made using file from Natural Earth (https://www.naturalearthdata.com/). The base map was made using files from The Geospatial Information Authority of Japan (https://www.gsi.go.jp/kiban/index.html).

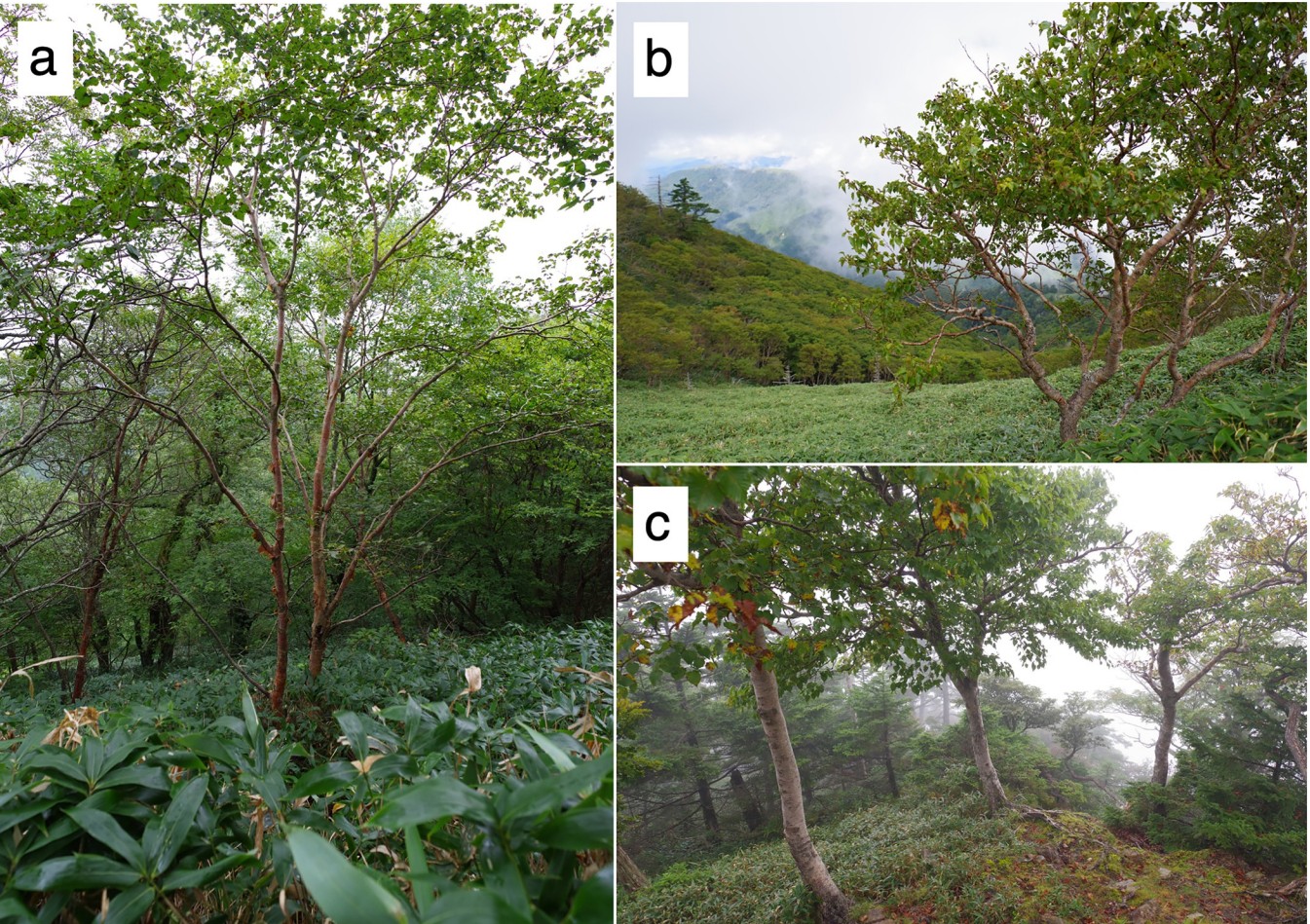

**Fig 2. Images of *Betula ermanii* in the southern boundary populations.** *Betula ermanii* habits in (a) Ishizuchi-san (ISZ) and (b) Tsurugi-san (TRG) on Shikoku Island and (c) Shaka-gatake (SHK) on the Kii Peninsula. These images were taken by the author T. A.

September 2022 (Table 2). Three early leaves were collected from 18 and 30 trees per population, and seeds from 3 to 14 trees per population, except for the YGT population (Table 2). For YGT, no seed material could be obtained, so subsequent analyses could only be performed on leaf samples.

All leaf and seed materials for the ploidy estimation and morphological comparison were permitted and/or approved by each of landowner and manager listed below. URU: Field Science Center for Northern Biosphere, Hokkaido University; HKD: Botanical Gardens, Tohoku University; YGT: Mountain Science Center, University of Tsukuba; BDS: Aizu National Forest District Office, Forestry Agency; SHK: Kinki Regional Environment Office, Ministry of the Environment; TRG: Tokushima National Forest District Office, Forestry Agency; ISZ: Tokushima National Forest District Office, Forestry Agency and Toyo Regional Office, Ehime Prefecture.

### Estimation of ploidy level

The DNA ploidy level of each sample was estimated using flow cytometry. An appropriate amount of fresh leaves was added to 50μl 4′,6-diamidino-2-phenylindole (DAPI) solution (supplemented with 2% polyvinylpyrrolidone K30 and 2μl/ml 2-mercaptoethanol) and co-

**Table 1. The location of the *Betula ermanii* populations and their estimated ploidy level.**

| Population | Lat | Long | Alt | *n* | Origin | Ploidy |
|---|---|---|---|---|---|---|
| URU | 44.39 | 142.28 | 486.70 | 2 | Common garden | 4× |
| AKS | 42.99 | 144.92 | 104.80 | 2 | Common garden | 4× |
| HKD | 40.65 | 140.85 | 898.10 | 2 | Common garden | 4× |
| GYS | 39.18 | 141.74 | 842.00 | 2 | Common garden | 4× |
| CKS | 39.07 | 140.04 | 1256.10 | 2 | Common garden | 4× |
| BDS | 37.63 | 140.05 | 1075.70 | 2 | Common garden | 4× |
| MKT | 36.77 | 138.82 | 1293.40 | 2 | Common garden | 4× |
| YGT | 35.92 | 138.50 | 1570.74 | 2 | Natural habitat | 4× |
| APW | 35.81 | 137.84 | 2456.90 | 2 | Common garden | 4× |
| NGH | 35.77 | 136.51 | 1495.40 | 2 | Common garden | 4× |
| APS | 35.14 | 138.05 | 1523.40 | 2 | Common garden | 4× |
| SHK | 34.11 | 135.90 | 1779.60 | 10 | Natural habitat | 2× |
| TRG | 33.86 | 134.09 | 1862.91 | 8 | Natural habitat | 2× |
| | | | | 2 | Natural habitat | 4× |
| ISZ | 33.77 | 133.13 | 1641.62 | 8 | Natural habitat | 2× |
| | | | | 2 | Natural habitat | 4× |

Lat, latitude (°N); Long, longitude (°E); Alt, altitude (m); *n*, number of individuals used to estimate the ploidy level. The 14 populations were: URU, Uryu; AKS, Akkeshi; HKD, Hakkoda; GYS, Goyo-san; CKS, Choukai-san; BDS, Bandai-san; MKT, Mikuni-touge; YGT, Yatsu-gatake; APW, Alps-west; NGH, Nougouhaku-san; APS, Alps-south; SHK, Shaka-gatake; TRG, Tsurugi-san; ISZ, Ishizuchi-san.

chopped with the internal reference standard *Epipremnum aureum*. The isolated nuclei were then stained with 950μl DAPI solution, and incubated for 15–30 min at room temperature. The samples were filtered through a 30μm nylon mesh to remove tissue debris, and the fluorescence intensity of 5000 particles was analyzed using a Quantum P Flow Cytometer (Quantum Analysis, Germany). The resulting histograms were analyzed with CyPAD software version 1.3 (Quantum Analysis, Germany). We initially set the peak position of the fluorescence histogram of *E. aureum* at 350 (relative value). At this setting, the peak position of the fluorescence histogram of tetraploid *B. ermanii* is 100, while that of diploid *B. ermanii* is 50. Then, we could estimate the ploidy level of each sample.

## Morphological comparison

A morphological comparison was made of the sampled leaves and seeds, as is commonly carried out for species and subspecies identification of *Betula* [9, 31]. The focus was the morphological traits of early leaves of *B. ermanii*, the first and second proximal leaves of a current-year shoot, because their phenotypes are more stable than those of later leaves, which are more strongly influenced by current-year conditions [32, 33]. Three early leaves from each tree were scanned using a flat-bed scanner CanoScan LiDE 400 (Canon, Japan), and the following leaf morphological traits measured using ImageJ software (https://imagej.nih.gov/ij/): leaf area (cm$^2$), leaf length (mm) (Fig 3A), leaf width (mm) (Fig 3B), leaf length/leaf width, perimeter of leaf (mm), perimeter of leaf/leaf area, the number of leaf veins, length of leaf base (mm) (Fig 3C), and length of leaf base/leaf length (Table 2). The number of leaf veins and length of leaf base were measured for one side of each leaf. Average trait values were used in the analyses.

To analyze the seed morphology, 100 seeds per tree, from 3 to 14 trees per population (except for the YGT population; Table 2) were air-dried and weighed using an M-power

**Table 2.** The number of trees per *Betula ermanii* population used for the morphometric analyses, and their trait values.

| | | URU | HKD | BDS | YGT | SHK | TRG | ISZ |
|---|---|---|---|---|---|---|---|---|
| Number of trees used for leaf morphology | | 30 | 19 | 24 | 18 | 30 | 30 | 30 |
| Number of trees used for seed morphology | | 8 | 3 | 7 | 0 | 14 | 6 | 11 |
| Leaf area (cm$^2$) | Mean | 42.01 | 34.82 | 29.88 | 29.91 | 22.21 | 17.97 | 23.72 |
| | SD | 14.10 | 9.89 | 7.37 | 4.72 | 4.37 | 3.86 | 6.36 |
| Leaf length (mm) | Mean | 86.43 | 79.58 | 72.59 | 77.21 | 62.72 | 75.34 | 69.95 |
| | SD | 15.02 | 15.82 | 13.75 | 14.11 | 12.70 | 12.24 | 15.72 |
| Leaf width (mm) | Mean | 53.58 | 49.79 | 38.41 | 40.03 | 41.97 | 30.77 | 39.17 |
| | SD | 13.76 | 12.69 | 9.04 | 7.63 | 6.43 | 5.31 | 7.59 |
| Leaf length/width | Mean | 1.70 | 1.69 | 1.96 | 2.00 | 1.54 | 2.51 | 1.86 |
| | SD | 0.46 | 0.52 | 0.49 | 0.52 | 0.47 | 0.50 | 0.60 |
| Leaf perimeter (mm) | Mean | 409.76 | 423.20 | 362.07 | 370.36 | 430.81 | 330.39 | 396.39 |
| | SD | 69.98 | 83.30 | 68.68 | 42.11 | 52.63 | 46.93 | 60.74 |
| Leaf perimeter/leaf area | Mean | 10.31 | 12.51 | 12.71 | 12.58 | 19.80 | 18.85 | 17.33 |
| | SD | 1.96 | 1.79 | 3.80 | 1.83 | 2.50 | 3.04 | 2.97 |
| Leaf veins (*n*) | Mean | 9.61 | 10.04 | 10.47 | 10.70 | 10.89 | 9.70 | 10.68 |
| | SD | 0.89 | 0.96 | 1.05 | 0.98 | 0.74 | 0.67 | 0.91 |
| Leaf base length (mm) | Mean | 0.53 | 2.10 | -1.15 | 2.24 | 4.98 | 4.13 | 5.15 |
| | SD | 4.14 | 2.64 | 2.67 | 2.08 | 1.44 | 1.27 | 1.76 |
| Leaf base length/leaf length | Mean | 0.01 | 0.03 | -0.02 | 0.03 | 0.08 | 0.06 | 0.07 |
| | SD | 0.05 | 0.03 | 0.04 | 0.03 | 0.02 | 0.02 | 0.02 |
| Seed weight (×100) (mg) | Mean | 89.46 | 48.97 | 44.57 | – | 40.14 | 45.07 | 36.52 |
| | SD | 20.72 | 3.38 | 7.08 | – | 8.64 | 6.47 | 10.39 |
| Seed area (×100) (cm$^2$) | Mean | 752.54 | 685.00 | 660.35 | – | 691.92 | 720.28 | 708.73 |
| | SD | 104.41 | 50.94 | 95.37 | – | 98.12 | 64.28 | 118.26 |
| Seed length (mm) | Mean | 2.62 | 2.48 | 2.39 | – | 2.43 | 2.61 | 2.59 |
| | SD | 0.28 | 0.18 | 0.24 | – | 0.19 | 0.12 | 0.28 |
| Seed width (mm) | Mean | 2.05 | 1.96 | 1.64 | – | 1.84 | 1.65 | 1.70 |
| | SD | 0.21 | 0.09 | 0.06 | – | 0.13 | 0.11 | 0.15 |
| Seed length/width | Mean | 1.29 | 1.26 | 1.46 | – | 1.32 | 1.59 | 1.53 |
| | SD | 0.10 | 0.04 | 0.15 | – | 0.08 | 0.07 | 0.15 |
| Seed wing length (mm) | Mean | 0.65 | 0.63 | 0.72 | – | 0.87 | 0.77 | 0.78 |
| | SD | 0.14 | 0.11 | 0.10 | – | 0.10 | 0.09 | 0.09 |
| Seed wing length /seed width | Mean | 0.32 | 0.32 | 0.44 | – | 0.47 | 0.47 | 0.46 |
| | SD | 0.07 | 0.06 | 0.06 | – | 0.05 | 12.24 | 15.72 |

The 7 populations were: URU, Uryu; HKD, Hakkoda; BDS, Bandai-san; YGT, Yatsu-gatake; SHK, Shaka-gatake; TRG, Tsurugi-san; ISZ, Ishizuchi-san

electronic scale (Sartorius, Germany), and scanned using the flat-bed scanner. The area of the 100 seeds (cm$^2$) was measured using ImageJ software. In addition, we measured the seed length (mm) (Fig 3D), seed width (mm) (Fig 3E) and length of the seed wing (mm) (Fig 3F) of 10 seeds per tree, and calculated the seed length/seed width and length of seed wing/seed width. Except for the weight and seed area, the average trait values of 10 seeds per tree were used in the analyses.

## Statistical analysis

All analyses were conducted in R 4.3.1 [34]. Leaf and seed morphological traits were compared between populations using a Tukey multiple comparison calculated by the R packages

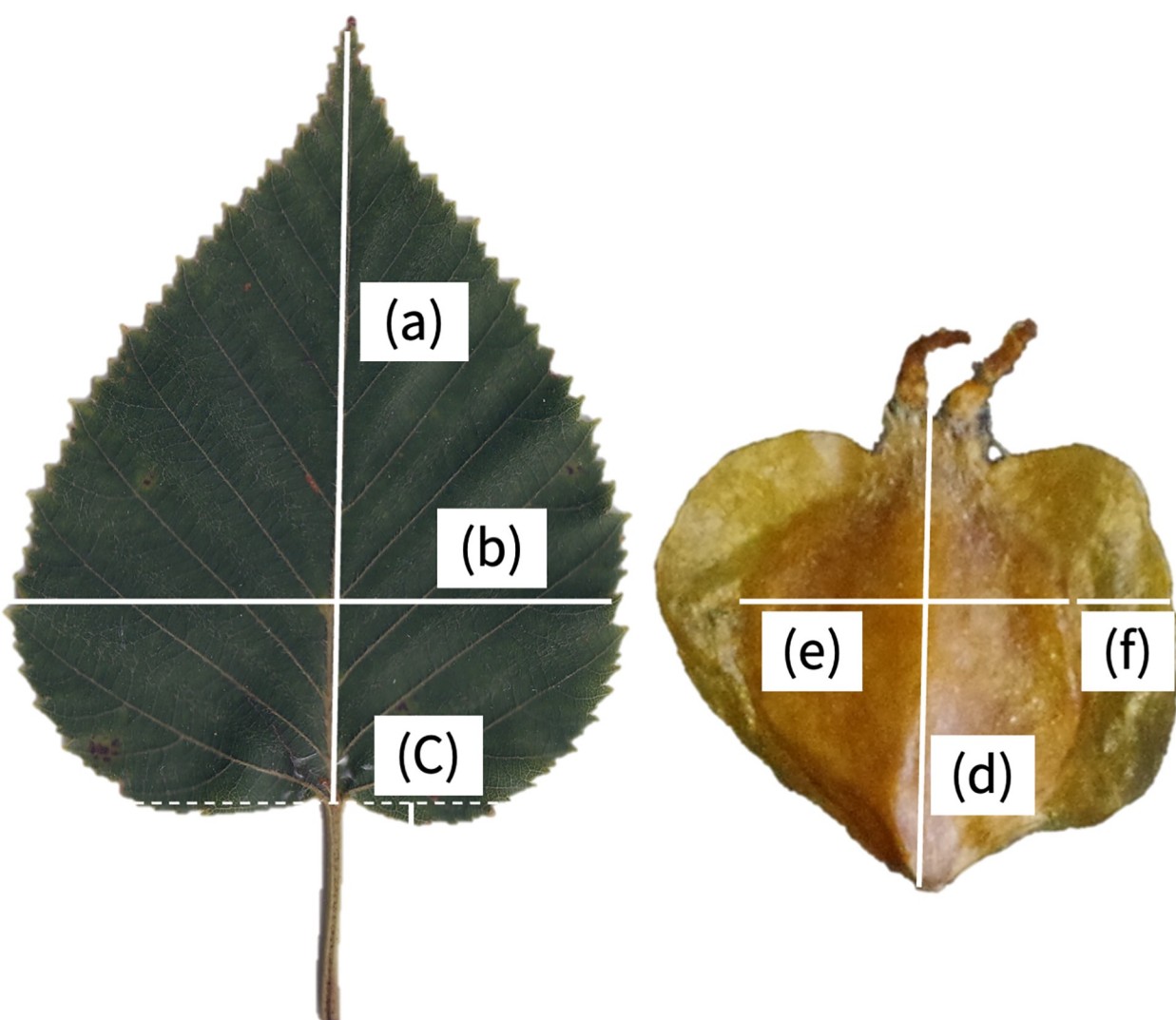

**Fig 3. Examples of leaf and seed images of *Betula ermanii* and the morphological measurements taken.** (a) Leaf length (mm); (b) Leaf width (mm); (c) Length of leaf base (mm); (d) Seed length (mm); (e) Seed width (mm); (f) Length of seed wing (mm). The leaf and seed images were taken by the author T. A.

'multcomp' and 'emmeans'. In addition, we performed a principal component analysis (PCA) on both the leaf morphological traits and seed morphological traits using the R function 'prcomp'. The differences in morphological traits between the populations and ploidy levels were tested using ANOVA with the R function 'aov'.

## Results

### Ploidy level

Flow cytometry analysis revealed that the ploidy level of all the examined individuals, apart from those from SHK, TRG and ISZ, were tetraploid (Fig 1; Table 1). In contrast, the ploidy level of all the individuals from SHK and most from TRG and ISZ were estimated to be diploid. Two individuals from lower altitudes of the TRG and ISZ populations were estimated to be tetraploid (Fig 1; S1 Table).

## Leaf morphology

We analyzed the leaf morphological traits of 181 individuals from seven *B. ermanii* populations (Table 2). The PCA results for leaf morphology showed that individuals from tetraploid (URU, HKD, BDS and YGT) and diploid (SHK, TRG and ISZ) populations formed clusters (Fig 4).

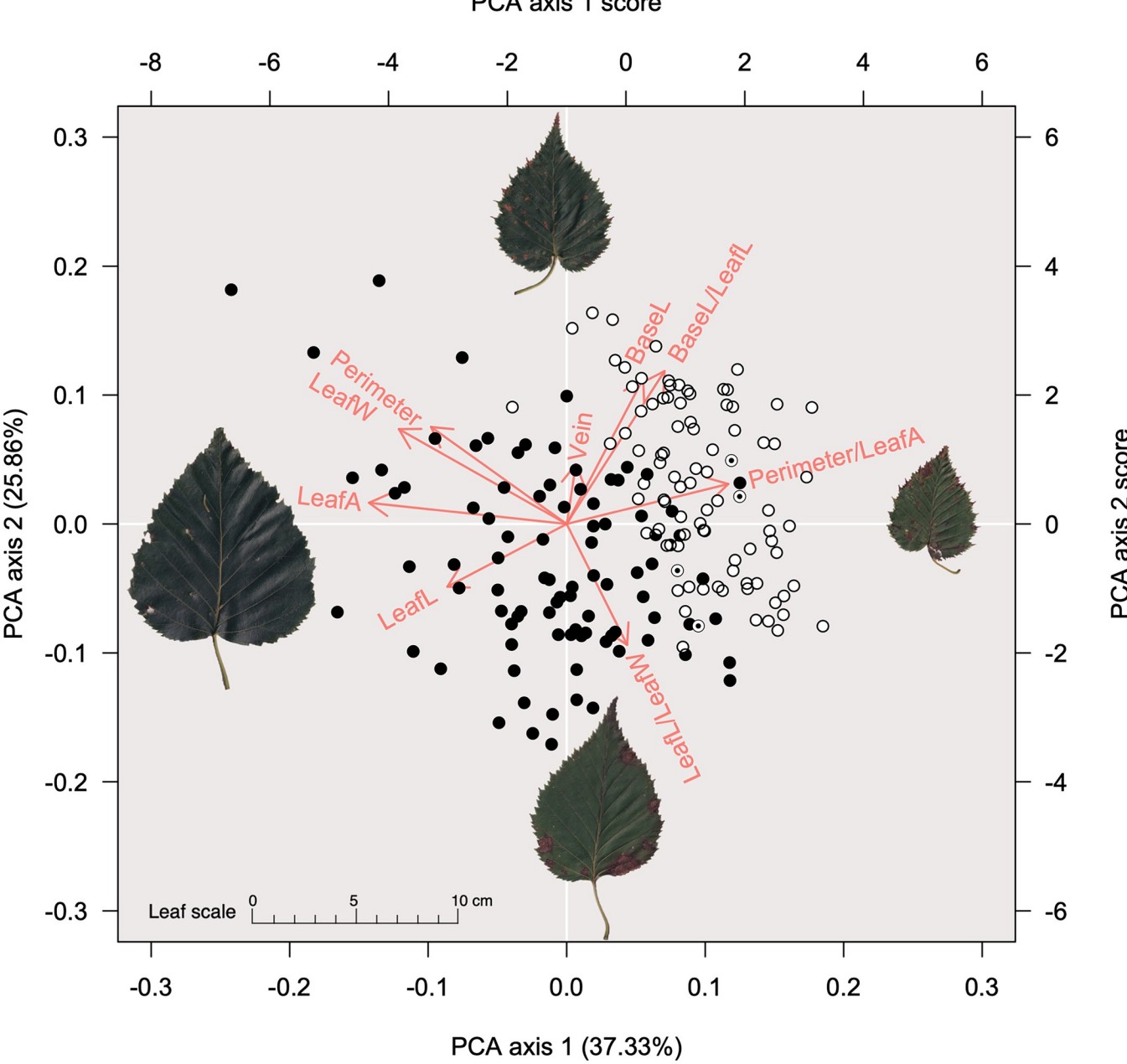

**Fig 4. A principal component analysis (PCA) of *Betula ermanii* leaf morphological traits.** Black circles indicate individuals from tetraploid populations; white circles indicate individuals from diploid populations; black-filled double circles indicate tetraploid individuals from TRG and ISZ populations. Each circle is based on the principal component score. Red letters and arrows indicate the principal component loadings for each variable (axes 1 and 2), calculated from the rotation and standard deviation of each variable. The images of leaves are typical silhouettes of relevant samples along each axis. LeafA, leaf area (cm$^2$); LeafL, leaf length (mm); LeafW, leaf width (mm); LeafL/LeafW, leaf length/leaf width; Perimeter, perimeter of leaf (mm); Perimeter/LeafA, perimeter of leaf/ leaf area; Vein, the number of leaf veins; BaseL, length of leaf base (mm); BaseL/LeafL, length of leaf base/leaf length. The leaf images were taken by the author T. A.

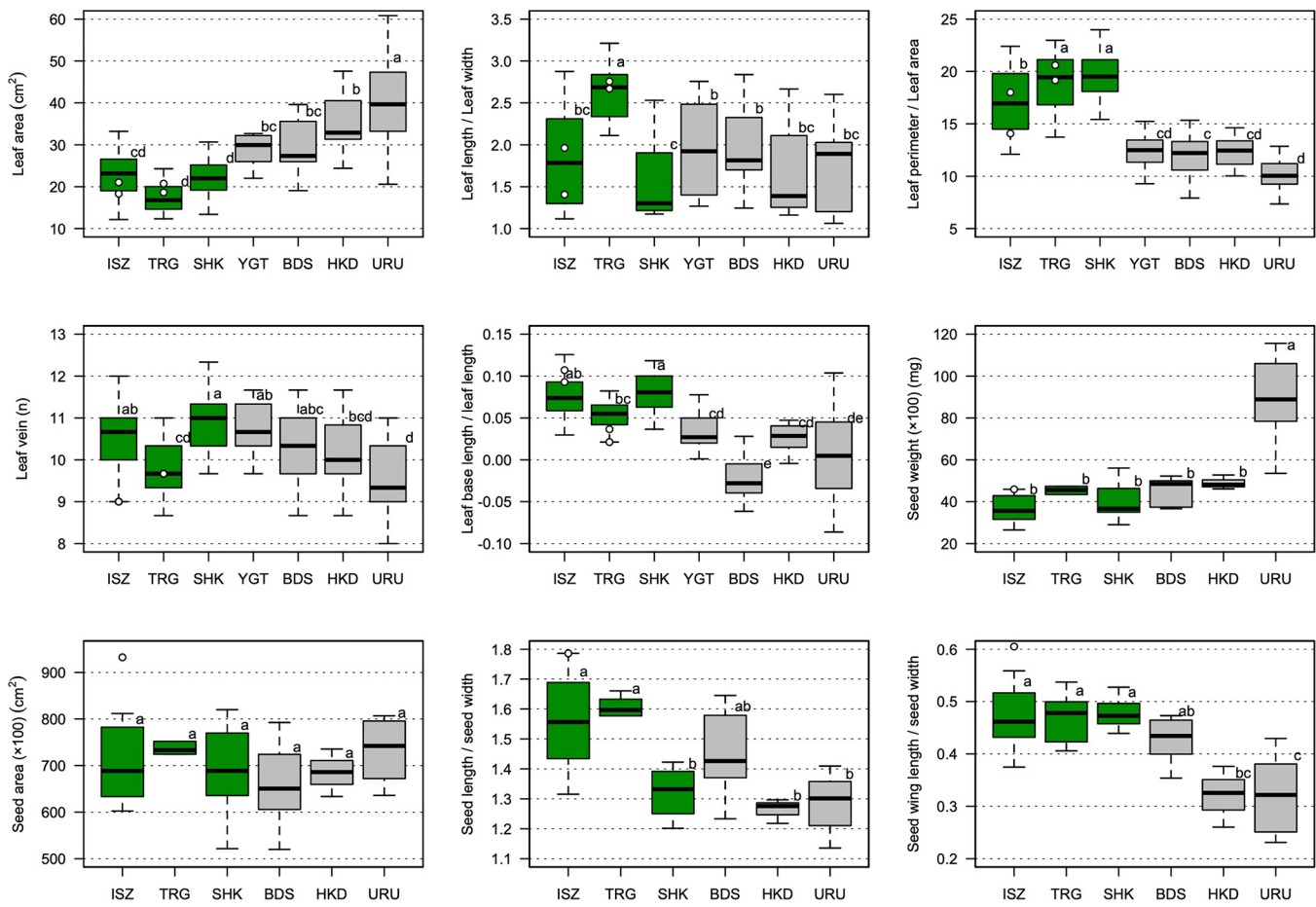

**Fig 5. Boxplots of leaf area, leaf length/leaf width, perimeter of leaf/leaf area, the number of leaf veins, length of leaf base/leaf length, weight of 100 seeds, area of 100 seeds, seed length/seed width, and length of seed wing/seed width.** Green plots indicate the traits of diploid populations; gray plots indicate the traits of tetraploid populations; white circles indicate the traits of tetraploid individuals from TRG and ISZ populations. For site abbreviations, see Fig 1; higher latitudinal populations are shown on the right-hand side, and lower latitudinal populations on the left. Different letters at the top of each box indicate statistically significant differences between populations based on a Tukey multiple comparison performed for each experimental year, based on a 95% confidence level.

Axis 1 explained 37.33% of the variance, which was associated with leaf area, leaf length, leaf width and length of perimeter/leaf area. Axis 2 explained 25.86% of the variance, which was associated with leaf length/leaf width, length of leaf base and length of leaf base/leaf length. Individuals from diploid populations fell on the higher side of axis 1 (Fig 4) and were characterized by a higher ratio of length of perimeter to leaf area and a higher ratio of length of leaf base to leaf length (Fig 5). The ANOVA results indicated that the tetraploid and diploid populations were separated by statistically significant differences in leaf area, leaf length, leaf width, length of perimeter/leaf area, number of leaf veins, length of leaf base, length of leaf base/leaf length, and axes 1 and 2 of the PCA (S2 Table). The tetraploid individuals in TRG and ISZ populations could not be differentiated from the diploid populations based on leaf morphology (Figs 4 and 5).

## Seed morphology

We analyzed the seed morphological traits of 49 individuals from six *B. ermanii* populations (Table 2). The PCA results of seed morphology showed that individuals from tetraploid and

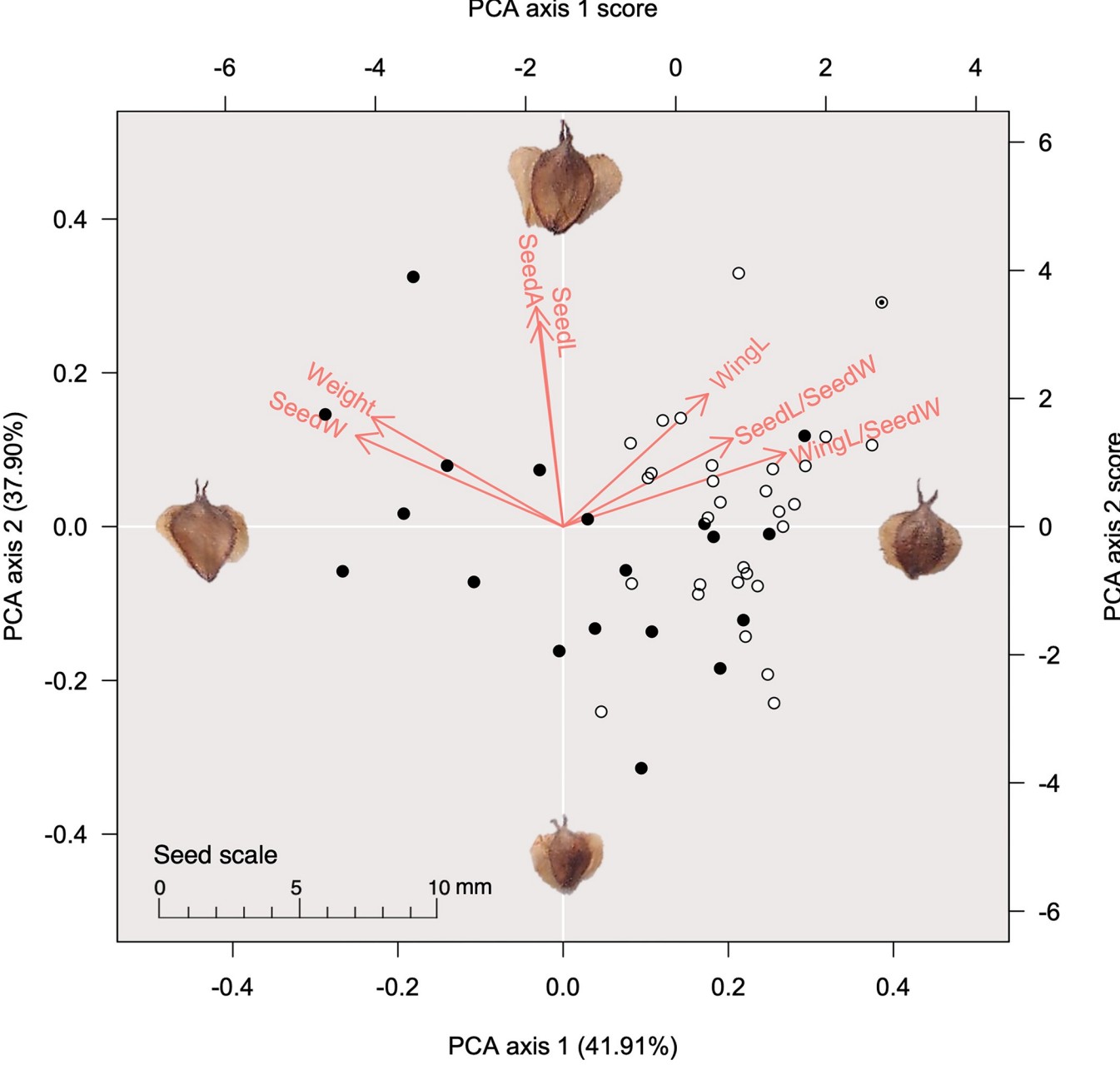

**Fig 6. A principal component analysis (PCA) of *Betula ermanii* seed morphological traits.** Black circles indicate individuals from tetraploid populations; white circles indicate individuals from diploid populations; black-filled double circles indicate tetraploid individuals from the ISZ population. Each circle is based on the principal component score. Red letters and arrows indicate the principal component loadings for each variable (axes 1 and 2), calculated from the rotation and standard deviation of each variable. The images of seeds are typical silhouettes of samples close to the values along each axis. SeedA, area of 100 seeds (cm$^2$); Weight, weight of 100 seeds (mg); SeedL, seed length (mm); SeedW, seed width (mm); SeedL/SeedW, seed length/seed width; WingL, length of seed wing (mm); WingL/SeedW, length of seed wing/seed length. The seed images were taken by the author T. A.

diploid populations formed clusters (Fig 6). Axis 1 explained 41.91% of the variance, which was associated with the weight of seeds and seed width. Axis 2 explained 37.90% of the variance, which was associated with seed area, seed length and length of seed wings. Individuals from diploid populations fell on the higher side of axis 1 (Fig 6) and were characterized by a higher ratio of seed wing length to seed width (Fig 5). The ANOVA results indicated that the

tetraploid and diploid populations were separated by statistically significant differences in seed weight, seed width, length of seed wing, length of seed wing/seed width, and axis 1 of the PCA (S3 Table). Tetraploid individuals in the ISZ population could not be separated from diploid populations based on seed morphology (Figs 5 and 6).

## Discussion

In a previous study, one of the southern boundary populations, SHK, was found to differ genetically from other *B. ermanii* populations, as indicated by a 0.507–0.688 ρ statistic [35], an analog of the commonly used $F_{ST}$ [26]. It also appeared to display inbreeding depression based on the value of relatedness, although inbreeding depression in polyploids is generally masked by their more frequent heterozygous alleles [2]. In the past, *B. ermanii* populations on Shikoku Island, including TRG and ISZ, have been described as a separate species, *B. shikokiana* NAKAI [21–23]. Corroborating these recorded differences, the current study shows that individuals in these populations present as diploid and have different leaf and seed morphologies compared with other *B. ermanii* populations (Figs 1, 4 and 6). Aside from one study that has reported an unusual chromosome number ($2n = 63$, compared with the usual $2n = 56$) for *B. ermanii* seedlings from Hokkaido [36], this study is the first to report diploid *B. ermanii*.

Based on climate-driven range dynamics, as temperature rise, populations of *B. ermanii* are likely to move to more northerly or higher altitude habitats, as will other tree species that are found in cool-temperate and alpine forests in Japan [37, 38]. The study populations of *B. ermanii* at the species' southern boundary represent its rear-edge populations in Japan. Rear-edge populations are typically restricted to a particular habitat, such as islands, within a matrix of unsuitable conditions, and are often small and isolated [39]. Some rear-edge populations of *Alnus gluticosa*, which is usually known as diploid, have been found to be tetraploid, and they appear to have emerged from auto-polyploidization and have persisted in long-term isolation as small populations [40, 41]. Similarly, the alpine shrub *Vaccinium vitis-idaea* can be tetraploid and self-compatible at lower altitudes, compared with its alpine populations, which are diploid and self-incompatible [42]. These studies imply that polyploid lineages can help maintain genetic variability and avoid inbreeding depression in small habitats with less suitable niche conditions [2]. In contrast, this current study found the opposite: diploid rear-edge small isolated populations, and tetraploid populations elsewhere. Molecular analyses suggest that *B. ermanii* is an allotetraploid species that arose via the interspecific hybridization of diploid progenitors [15]. Therefore, we suggest that the diploid *B. ermanii* rear-edge populations are relict lineages, and there is the possibility that they are one of the diploid progenitors of the tetraploid *B. ermanii*. Many relict endemic species occur in regions with rear-edge populations, represented in this study by the Kii Peninsula and Shikoku, where they are called 'Sohayaki elements' [27] and are thought to represent the most ancient plant groups in Japan [43]. Some coniferous species of higher altitudes, such as *Abies homolepis*, *Abies veitchii*, *Picea jezoensis* and *Thuja standishii*, also have distinctive genetic features in these areas [28–30], which is consistent with the suggestion that the relict ancient lineage of *B. ermanii* is represented by current populations.

Previous study suggested that one of the diploid progenitors of *B. ermanii* was *B. costata* [15]. Disjunct distributions of *B. costata* across Japan have been reported only recently, mainly because its bark is morphologically similar to *B. ermanii* and its range of distribution is overlapped with *B. ermanii* in central Japan [18, 19]. However, we did not recognize the diploid *B. ermanii* found at the southern boundary as *B. costata* because the diploid *B. ermanii* has some morphological differences compared with *B. costata*, for example the number of leaf veins is around 14 in *B. costata* [44], while the diploid *B. ermanii* has around 10 (Table 2; Fig 5). In

addition, the texture of the bark is different between these two species [18] (Fig 2). A follow-up study is needed to understand their phylogenetic relationships in more detail, based on genetic analyses across Japan.

The leaves of the diploid *B. ermanii* had a larger length of perimeter/leaf area and larger length of leaf base (Figs 4 and 5) compared with tetraploid individuals, which had a heart-shaped leaf base and many larger leaf teeth. This suggests that diploid and tetraploid individuals can be differentiated morphologically. Leaf specimens in the Natural Science Museum of Japan labelled *B. shikokiana* NAKAI, which now combined as *B. ermanii*, display very similar morphological features to our diploid samples. However, tetraploid individuals from the TRG and ISZ populations did not show a distinctive morphology compared with diploid individuals (Figs 4–6). As this study only included a few tetraploid individuals from the TRG and ISZ populations, further research based on more samples is required.

Diploid and polyploid individuals of some species display differences in traits such as leaf anatomy and transpiration rate [1]. Polyploids have a larger cell size than diploids, because of whole-genome duplication, hence they also have larger seeds and greater resistance to water stress [7, 8]. For example, [45] have demonstrated that polyploid *Betula* displays more stability in xylem hydraulics than diploid *Betula*, and is found in more stressful habitats. In *B. papyrifera*, higher ploidy individuals have been reported to display better water deficit tolerance because of their larger stomatal size, and are found in drier areas [13]. Differences in traits between tetraploid and diploid individuals were also observed in *B. ermanii* (Figs 4 and 6), with the differences in ploidy probably contributing to the differences in morphology. Similarly, the geographic range of diploid and tetraploid *B. ermanii* appears to be separate (Fig 1), and these two lineages probably occupy different niches.

Consequently, the southern boundary populations of *B. ermanii* on Shikoku and the Kii Peninsula show a distinctive ploidy and morphology compared with tetraploid *B. ermanii*, and a taxonomic revision is deemed necessary. In this context, the diploid lineages of two tetraploid *Betula* populations have been described as new species [17, 46]. However, mixed-ploidy species, taxa containing at least two ploidies and sometimes odd-number ploidy individuals that originate from hybrids [47, 48], have been reported in *Betula*: *B. papyrifera* (2×, 3×, 5× and 6×) [13], *B. pendula* (2× and 4×) [14], *B. chinensis* (6× and 8×) [9] and *B. dahurica* (6× and 8×) [12]. We found diploid and tetraploid *B. ermanii* individuals in both TRG and ISZ (Fig 1), and tetraploid individuals in these populations were only located at lower elevations (1562, 1573, 1465 and 1563 m a.s.l.; S1 Table), closer to the car parking area at trail entrances and along the trails. We presume that these trees were not intentionally planted recently, because their girth at breast height (GBH) was not significantly lower than other individuals (S1 Table). In order to review their taxonomy in depth, the phylogenetic relationships and past population dynamics of Japanese *B. ermanii* populations need to be investigated further using genetic analyses.

## Conclusions

*Betula ermanii* has long been known as a tetraploid. Unexpectedly, most *B. ermanii* individuals located at its southern boundary populations appeared to be diploid. Leaf and seed morphologies exhibited notable distinctions between tetraploids and diploids. Diploid individuals were characterized by leaves featuring a heart-shaped base and numerous leaf teeth, and seeds with relatively longer wings. The diploid populations could be regarded as a cryptic relict lineage of *B. ermanii*, suggesting a potential ancestral diploid lineage of *B. ermanii* and a relict plant population that have been introduced to Japan in ancient times. A comprehensive examination of those *Betula* phylogenetic relationships would facilitate an informed discussion of taxonomic revisions.

## Supporting information

**S1 Table. Location, tree height, girth, SLA and ploidy for examined individuals in *Betula ermanii* southern boundary populations.**
(XLSX)

**S2 Table. Results of ANOVAs of the leaf morphological traits and PCA axes.**
(XLSX)

**S3 Table. Results of ANOVAs of the seed morphological traits and PCA axes.**
(XLSX)

**S1 Data. Raw data for the leaf and seed morphological comparisons.**
(CSV)

## Acknowledgments

We thank the staff of the Nayoro Research Office, Hokkaido University, the Yatsugatake Forest Station, University of Tsukuba, and Mt Hakkoda Botanical Garden, Tohoku University, for their assistance with the field surveys. We also thank K. Watanabe, J. Iijima, K. Uchiyama and T. Nagamitsu for providing helpful suggestions, and S. Hayakashi and C. Yihan for helping with sample collection.

## Author Contributions

**Conceptualization:** Takaki Aihara.

**Formal analysis:** Takaki Aihara.

**Funding acquisition:** Takaki Aihara, Yoshihiko Tsumura.

**Investigation:** Takaki Aihara, Kyoko Araki, Yoshihiko Tsumura.

**Project administration:** Yoshihiko Tsumura.

**Visualization:** Takaki Aihara.

**Writing – original draft:** Takaki Aihara.

**Writing – review & editing:** Yoshihiko Tsumura.

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
