## [Decision Letter · Decision Letter 0]

22 Mar 2024

PONE-D-24-05878Cryptic diploid lineage of Betula ermanii at its southern boundary populations in JapanPLOS ONE

Dear Dr. Tsumura,

Thank you for submitting your manuscript to PLOS ONE. After careful consideration, we feel that it has merit but does not fully meet PLOS ONE’s publication criteria as it currently stands. Therefore, we invite you to submit a revised version of the manuscript that addresses the points raised during the review process.

We look forward to receiving your revised manuscript.

Kind regards,

Vikas Sharma, Ph.D

Academic Editor

PLOS ONE

Journal Requirements:

"This work was supported by: the JSPS KAKENHI program, Recipient: YT, Grant Number: 21H04732, URL: https://www.jsps.go.jp/j-grantsinaid/; JST Next Generation Researchers Challenging Research program, Recipient: TA, Grant Number: JPMJSP2124, URL: https://www.jst.go.jp/jisedai/"

7. We note that Figure 1 in your submission contain map images which may be copyrighted. All PLOS content is published under the Creative Commons Attribution License (CC BY 4.0), which means that the manuscript, images, and Supporting Information files will be freely available online, and any third party is permitted to access, download, copy, distribute, and use these materials in any way, even commercially, with proper attribution. For these reasons, we cannot publish previously copyrighted maps or satellite images created using proprietary data, such as Google software (Google Maps, Street View, and Earth). For more information, see our copyright guidelines: http://journals.plos.org/plosone/s/licenses-and-copyright.

(1) You may seek permission from the original copyright holder of Figure 1 to publish the content specifically under the CC BY 4.0 license.  

**Additional Editor Comments:**

Authors are requested to go through the comments of reviewers and incorporate changes / modification in manuscript accordingly.

Reviewers' comments:

Reviewer's Responses to Questions

**Comments to the Author**

1. Is the manuscript technically sound, and do the data support the conclusions?

Reviewer #1: Yes

Reviewer #2: Yes

2. Has the statistical analysis been performed appropriately and rigorously? 

Reviewer #1: Yes

Reviewer #2: Yes

3. Have the authors made all data underlying the findings in their manuscript fully available?

Reviewer #1: Yes

Reviewer #2: Yes

4. Is the manuscript presented in an intelligible fashion and written in standard English?

Reviewer #1: Yes

Reviewer #2: Yes

5. Review Comments to the Author

Reviewer #1: The research manuscript entitled "Cryptic Diploid Lineages of Betula at its Southern Boundary Populations in Japan" delves into Betula species' previously undiscovered terrain, exploring diploid lineages at the species' southern limits in Japan. The work illuminates Betula populations' hidden diversity and sheds light on this ecologically important genus' evolutionary history.

However, the current work lacks thoroughness and significance in this research due to some flaws. The validity and trustworthiness of the results are questioned due to insufficient sample size. The paper's results are presented in a confusing way, and the interpretations of these results are strongly conjectural. Some of these are mentioned here:

Line 78: Authors have discussed Betula species, but not clear which species authors are talking about.

Line 80: How same species as been named differently in different regions? What is the acceptable botanical name of the species?

Line 87: Authors are talking about which species?

Line 285: Authors stated that B. costata is diploid progenitor of B. ermanii. But, in next lines 286-87, authors are confirming that both are same species. And later on, authors also mentioning inadequate study related to this.

Further, Line 295-297: Authors agreed to the morphological similarity of their diploid specimen to the B. shikokiana, which is confusing, weather both are similar species or different.

Finally, the results are not clear enough to draw any firm conclusions from the study, which casts doubt on the writers' expertise in the field. The data presented in the introduction and discussions appears to be comparable and requires correction.

To summary, this study has had a greater impact on our understanding of cryptic diploid lineages in Betula populations, but manuscript needed to revise with prudence until these core concerns are resolved.

Reviewer #2: Abstract

The abstract provides a concise summary of the study, highlighting the focus on Betula ermanii populations in southern Japan and their morphological and ploidy variations. It effectively outlines the methodology and key findings, setting clear expectations for readers.

Introduction

The introduction provides a thorough background on polyploidy and its role in species diversification, effectively setting the stage for the study. It discusses the relevance of polyploidy to Betula species and outlines the uncertainties in Betula phylogenetics due to cryptic lineages. However, the introduction could be strengthened by providing more context on why Betula ermanii was chosen as the focus of the study and why investigating ploidy levels in this species is important

Materials and Methods

The section on study area and sampling is comprehensive, providing detailed information about the geographic locations of the study populations and the sampling methods employed. The use of common garden populations for comparison enhances the robustness of the study design.

The description of the estimation of ploidy level using flow cytometry is clear and replicable. However, it would be helpful to include more information about the specific protocols followed for DNA extraction and flow cytometry analysis to ensure transparency and reproducibility.

The morphological comparison methodology is well-described, particularly the use of leaf and seed traits for analysis. The inclusion of images (Fig. 2 and Fig. 3) enhances the clarity of the methods section.

Results

The results section presents the findings in a logical sequence, starting with the ploidy level estimation followed by morphological comparisons of leaves and seeds. The use of figures (Fig. 4, Fig. 5, and Fig. 6) aids in visualizing the data and understanding the patterns observed.

The identification of diploid and tetraploid populations based on ploidy level estimation is a significant finding, demonstrating variation within Betula ermanii populations. The morphological comparisons reveal distinct differences between diploid and tetraploid populations, particularly in leaf and seed traits.

Discussion

The discussion section effectively synthesizes the study's findings, contextualizes them within existing research, and offers valuable insights into the genetic and morphological diversity of Betula ermanii populations. With some enhancements to depth and integration, the discussion could further elucidate the ecological and evolutionary significance of the observed patterns and provide more concrete recommendations for future research and conservation efforts.

6. PLOS authors have the option to publish the peer review history of their article (what does this mean?). If published, this will include your full peer review and any attached files.

Reviewer #1: No

Reviewer #2: **Yes: **Saleem Wani

---

## [Author Response · Author response to Decision Letter 0]

8 May 2024

Responds to Academic Editor:

1. Our revised manuscript was made according to PLOS ONE's style requirements.

2. We have added the ethics statement and the full name of the authority that approved our sample collection in Methods section.

3. When we submit, we will provide the correct grant numbers for awards we received for this study in ‘Funding Information’.

5. We have provided an ORCID iD for the corresponding author.

6. We have added the ethics statement and the full name of the authority that approved our sample collection in Methods section in L 150-156.

7. The shapefiles of country boundary were made using “TM World Borders Dataset 0.3” (https://larmarange.github.io/prevR/reference/TMWorldBorders.html). The base map was made using files from The Geospatial Information Authority of Japan (https://www.gsi.go.jp/kiban/index.html). We have added this description in the caption of Fig 1. These files allow unrestricted use and distribution under the CC BY 4.0 licence. Then, we did not need any permission for using those in Fig 1.

Responds to Reviewer #1:

We greatly appreciate your extensive time and effort in providing the meaningful suggestions needed to publish these findings. As your comments, number of samplings for ploidy estimation except southern boundary populations is very small (only two individuals for each population). Based on the RAD-seq analysis (Aihara et al. 2023), it was observed that SHK, one of the southern boundary populations, could potentially be diploid. To confirm this observation, we conducted flow cytometry. Consequently, with the exception of three southern boundary populations, we sampled two samples from each population as controls. We have added such description in Materials L124-127. Finding of cryptic diploid individuals in B. ermanii is first report, and we believe that our manuscript has great impact and many readers show much attention.

Line 78: Authors have discussed Betula species, but not clear which species authors are talking about.

REPLY: Thank you for the comment. We revised the wording ‘related species’ to ‘separate species’ to clarify meaning in L80.

Line 80: How same species as been named differently in different regions? What is the acceptable botanical name of the species?

REPLY: Thank you for the comment. As many plant taxa, B. ermanii named differently in different regions by local botanist and researcher, and now combined as one species. We have added acceptable Japanese botanical name of the species in L80.

Line 87: Authors are talking about which species?

REPLY: We have added species name in L90.

Line 285: Authors stated that B. costata is diploid progenitor of B. ermanii. But, in next lines 286-87, authors are confirming that both are same species. And later on, authors also mentioning inadequate study related to this.

REPLY: Thank you for pointing it out. Betula ermanii and B. costata is separate species. We have revised sentences to clarify what we mean in L304-306.

Line 295-297: Authors agreed to the morphological similarity of their diploid specimen to the B. shikokiana, which is confusing, weather both are similar species or different.

REPLY: Thank you for the comment. Now B. shikokiana combined as B. ermanii. We have revised the sentence to clarify that.

Finally, the results are not clear enough to draw any firm conclusions from the study, which casts doubt on the writers' expertise in the field.

REPLY: As your comments, our conclusions were vague in the manuscript of previous version. Then, we have added new section “Conclusions” to clarify the conclusions and what we found in this study.

Responds to Reviewer #2:

We appreciate your comments on every part of the manuscript. These significantly improved the manuscript. We attended to all suggestions like below.

However, the introduction could be strengthened by providing more context on why Betula ermanii was chosen as the focus of the study and why investigating ploidy levels in this species is important

REPLY: As your comment, we have added sentences providing our insights why B. ermanii was chosen and why ploidy levels of this species should be investigated in L76-79.

However, it would be helpful to include more information about the specific protocols followed for DNA extraction and flow cytometry analysis to ensure transparency and reproducibility.

REPLY: Thank you for the comment. As your comment, we added more specific explanation about fluorescence analysis in L166-169. Actually, we did not extract DNA but nuclei. Then, we revised the wording to clarify it in L162.

With some enhancements to depth and integration, the discussion could further elucidate the ecological and evolutionary significance of the observed patterns and provide more concrete recommendations for future research and conservation efforts.

REPLY: As your comments, our discussion in the manuscript of previous version needs enhancement of what we found. Then, we have added new section “Conclusions” to clarify the conclusions and what we found in this study.

---

## [Decision Letter · Decision Letter 1]

28 Jun 2024

Cryptic diploid lineage of Betula ermanii at its southern boundary populations in Japan

PONE-D-24-05878R1

Dear Dr. Tsumura,

We’re pleased to inform you that your manuscript has been judged scientifically suitable for publication and will be formally accepted for publication once it meets all outstanding technical requirements.

Kind regards,

Vikas Sharma, Ph.D

Academic Editor

PLOS ONE

Additional Editor Comments (optional):

The authors have revised the manuscript as per suggestions of the reviewers. The manuscript can be accepted now.

Reviewers' comments:

Reviewer's Responses to Questions

**Comments to the Author**

1. If the authors have adequately addressed your comments raised in a previous round of review and you feel that this manuscript is now acceptable for publication, you may indicate that here to bypass the “Comments to the Author” section, enter your conflict of interest statement in the “Confidential to Editor” section, and submit your "Accept" recommendation.

Reviewer #2: All comments have been addressed

2. Is the manuscript technically sound, and do the data support the conclusions?

Reviewer #2: Yes

3. Has the statistical analysis been performed appropriately and rigorously? 

Reviewer #2: Yes

4. Have the authors made all data underlying the findings in their manuscript fully available?

Reviewer #2: Yes

5. Is the manuscript presented in an intelligible fashion and written in standard English?

Reviewer #2: Yes

6. Review Comments to the Author

Reviewer #2: the manuscript shows promise in contributing to the understanding of Betula ermanii populations in Japan. By addressing the mentioned points and refining the presentation of the research findings, the study can make a valuable addition to the scientific literature on plant genetics and evolutionary biology.

7. PLOS authors have the option to publish the peer review history of their article (what does this mean?). If published, this will include your full peer review and any attached files.

Reviewer #2: **Yes: **Mohd Saleem Wani

---

## [Editor Report · Acceptance letter]

4 Jul 2024

PONE-D-24-05878R1 

PLOS ONE

Dear Dr. Tsumura, 

I'm pleased to inform you that your manuscript has been deemed suitable for publication in PLOS ONE. Congratulations! Your manuscript is now being handed over to our production team.

Kind regards, 

on behalf of

Dr. Vikas Sharma 

Academic Editor

PLOS ONE